# Growth Developmental Defects of Mitochondrial Iron Transporter 1 and 2 Mutants in Arabidopsis in Iron Sufficient Conditions

**DOI:** 10.3390/plants12051176

**Published:** 2023-03-04

**Authors:** Joaquín Vargas, Isabel Gómez, Elena A. Vidal, Chun Pong Lee, A. Harvey Millar, Xavier Jordana, Hannetz Roschzttardtz

**Affiliations:** 1Departamento de Genética Molecular y Microbiología, Facultad de Ciencias Biológicas, Pontificia Universidad Católica de Chile, Santiago 8331150, Chile; 2ANID-Millennium Science Initiative Program-Millennium Institute for Integrative Biology (iBio), Santiago 8331150, Chile; 3Centro de Genómica y Bioinformática, Facultad de Ciencias, Ingeniería y Tecnología, Universidad Mayor, Santiago 8580745, Chile; 4Escuela de Biotecnología, Facultad de Ciencias, Ingeniería y Tecnología, Universidad Mayor, Santiago 8580745, Chile; 5ARC Centre of Excellence in Plant Energy Biology, School of Molecular Sciences, The University of Western Australia, Bayliss Building M316, Crawley, WA 6009, Australia

**Keywords:** mitochondria, iron transporters, MIT, developmental defects, RNA-seq

## Abstract

Iron is the most abundant micronutrient in plant mitochondria, and it has a crucial role in biochemical reactions involving electron transfer. It has been described in *Oryza sativa* that *Mitochondrial Iron Transporter* (*MIT*) is an essential gene and that knockdown mutant rice plants have a decreased amount of iron in their mitochondria, strongly suggesting that OsMIT is involved in mitochondrial iron uptake. In *Arabidopsis thaliana*, two genes encode MIT homologues. In this study, we analyzed different *AtMIT1* and *AtMIT2* mutant alleles, and no phenotypic defects were observed in individual mutant plants grown in normal conditions, confirming that neither *AtMIT1* nor *AtMIT2* are individually essential. When we generated crosses between the *Atmit1* and *Atmit2* alleles, we were able to isolate homozygous double mutant plants. Interestingly, homozygous double mutant plants were obtained only when mutant alleles of *Atmit2* with the T-DNA insertion in the intron region were used for crossings, and in these cases, a correctly spliced *AtMIT2* mRNA was generated, although at a low level. *Atmit1 Atmit2* double homozygous mutant plants, knockout for *AtMIT1* and knockdown for *AtMIT2*, were grown and characterized in iron-sufficient conditions. Pleiotropic developmental defects were observed, including abnormal seeds, an increased number of cotyledons, a slow growth rate, pinoid stems, defects in flower structures, and reduced seed set. A RNA-Seq study was performed, and we could identify more than 760 genes differentially expressed in *Atmit1 Atmit2*. Our results show that *Atmit1 Atmit2* double homozygous mutant plants misregulate genes involved in iron transport, coumarin metabolism, hormone metabolism, root development, and stress-related response. The phenotypes observed, such as pinoid stems and fused cotyledons, in *Atmit1 Atmit2* double homozygous mutant plants may suggest defects in auxin homeostasis. Unexpectedly, we observed a possible phenomenon of T-DNA suppression in the next generation of *Atmit1 Atmit2* double homozygous mutant plants, correlating with increased splicing of the A*tMIT2* intron containing the T-DNA and the suppression of the phenotypes observed in the first generation of the double mutant plants. In these plants with a suppressed phenotype, no differences were observed in the oxygen consumption rate of isolated mitochondria; however, the molecular analysis of gene expression markers, *AOX1a*, *UPOX*, and *MSM1*, for mitochondrial and oxidative stress showed that these plants express a degree of mitochondrial perturbation. Finally, we could establish by a targeted proteomic analysis that a protein level of 30% of MIT2, in the absence of MIT1, is enough for normal plant growth under iron-sufficient conditions.

## 1. Introduction

Iron is an essential nutrient, and it is well known that it is an integral constituent of many metalloproteins, primarily as part of heme groups and iron-sulfur clusters. As such, iron is essential for oxygen transport, electron transfer (redox), and catalytic reactions [1]. The biological versatility of iron is based on its capacity to be coordinated by proteins and to act as an electron donor and acceptor. Thus, iron can readily convert between its two common oxidation states, Fe^2+^ and Fe^3+^, by the loss or gain of one electron. However, iron is also potentially toxic due to its redox reactivity. Indeed, free iron acts as a catalyst for oxidative stress via Fenton reactions, which yield hazardous radicals with the capacity to attack cellular macromolecules and cause tissue damage. Consequently, a tight control of iron homeostasis is imperative to satisfy metabolic needs for iron and prevent the accumulation of toxic iron concentrations. Iron homeostasis involves all the processes that regulate the balance between iron uptake, its intracellular storage, and utilization [2].

In soil, Fe^2+^ undergoes spontaneous aerobic oxidation to Fe^3+^, which is virtually insoluble at physiological pH. This makes the acquisition of iron by cells and organisms challenging, despite its high abundance. The mechanism of iron uptake in the roots of *Arabidopsis thaliana* is now well-described and involves an acidification reduction-transport mechanism [3]. Under iron deficiency, ferric chelates are solubilized by local rhizosphere acidification caused by the release of protons by the Arabidopsis Plasma Membrane H^+^-ATPase2 (AHA2; [4]). Solubilized Fe^3+^ ions are then reduced to Fe^2+^ by the Reductase Ferric Reduction Oxidase2 (FRO2) [5] and, finally, transported into the cell by the Iron Transporter Iron Regulated Transporter1 (IRT1; [6,7]). The mechanisms governing the distribution of iron to specific organs, cells, and organelles are still very poorly understood.

In plants, in addition to its role in the mitochondrial electron transport chain, common to eukaryotes, iron is essential for chloroplast photosynthesis, as shown by the chlorosis of plants grown under iron-deficient conditions [8,9,10]. A total of twenty-two iron atoms are required per photosynthetic electron transport chain [11]. Iron import into the chloroplast was proposed to be performed by the Permease in Chloroplast1 (PIC1) localized in the inner envelope of this organelle [12]. *PIC1* knockout mutations result in dwarf plants with altered iron homeostasis. Before being transported into the chloroplast, iron is thought to be first reduced by the Ferric Reductase7 (AtFRO7), also localized in the chloroplast envelope [13]. Indeed, chloroplasts isolated from At*fro7* loss-of-function mutants have significantly reduced iron content and altered photosynthetic complexes [13]. Iron remobilization from leaf chloroplasts seems to be mediated by YSL4 and YSL6 [14]. Recently, it has been suggested that FPN3 has a role in the iron export from Arabidopsis chloroplasts and mitochondria [15].

In plant mitochondria, iron is more abundant than other transition metals such as Cu, Zn, and Mn, consistent with its crucial role as a component in electron transfer reactions [16]. It has been suggested that iron is transported to the mitochondria through the outer membrane by voltage-dependent anion protein channels (VDACs) and then to the mitochondrial matrix by the Mitochondrial Carrier Family (MCF) transporters. The MCF gene family, with more than fifty members in *Arabidopsis thaliana*, encodes membrane proteins containing six transmembrane domains [17]. Mitochondrial Iron Transporter (MIT), a member of the MCF in *Oryza sativa*, was the first mitochondrial iron transporter identified in plants [18]. Complementation studies using OsMIT demonstrated that it is able to transport iron into yeast mitochondria, and its function is essential in *Oryza sativa*. Knockdown plants for *OsMIT* showed a decrease in iron content in mitochondria and in aconitase activity, an iron-sulfur protein [18]. In *Arabidopsis thaliana*, it has been described that two genes encode MIT proteins [19]. Fusions with fluorescent proteins demonstrated that *AtMIT1* and *AtMIT2* localized to mitochondria, and plants knockout for *MIT1* (homozygous mutant) and knockdown for *MIT2* (heterozygous mutant) showed mitochondrial defects when plants were grown in iron deficiency conditions [19]. Both mitochondrial iron deficiency and excess seem to provoke oxidative stress in a mammalian model [20]. In plants, mitochondrial iron deficiency or excess also affects mitochondrial function [15,21], indicating that plant mitochondria have a crucial role in cellular metal homeostasis [22].

In this article, we characterize *Atmit1 Atmit2* double mutant plants (knockout for *MIT1* and knockdown for *MIT2*, using the same alleles used by Jain et al., 2019 [19]) grown in iron-sufficient conditions. These plants showed pleiotropic developmental defects, some of which strikingly resemble those found in auxin transport and sensing mutants. Transcriptomic data revealed a misregulation of genes involved in iron acquisition, synthesis of coumarins, formation of the Casparian strip, suberization, and root hair development. Furthermore, we demonstrate unambiguously, by crossing knockout mutants for MIT1 and MIT2, that MIT function is essential in Arabidopsis.

## 2. Results

Isolation of mutants in the two Arabidopsis genes encoding mitochondrial iron transporters (MIT).

In Arabidopsis, two genes (At1g07030 and At2g30160) encode proteins with high similarity to the rice Mitochondrial Iron Transporter (MIT) [18,23], and have recently been characterized [19]. The Arabidopsis MIT isoforms share 82% of their peptide sequence identity and similarity, including the putative mitochondrial targeting peptide. Both are 66–67% identical (77–78% similar) to the rice MIT protein (excluding the putative mitochondrial targeting peptides). To evaluate the potential role of the Arabidopsis proteins as mitochondrial iron transporters, At1g07030 (*MIT2*) and At2g30160 (*MIT1*) were used to transform the *MRS3-MRS4* knockout yeast (Δ*mrs3*Δ*mrs4*). *Saccharomyces cerevisiae* Mrs3 and Mrs4 are members of the MCF responsible for transporting Fe into mitochondria under low-Fe conditions, and the double knockout Δ*mrs3*Δ*mrs4* mutant grows poorly when Fe availability is low [24,25]. Each Arabidopsis MIT isoform was able to complement the growth defect of Δ*mrs3*Δ*mrs4* yeast cells (Appendix A), indicating that they can act as mitochondrial iron carriers. Our results are largely in agreement with previous observations by [19], with the exception of the lack of evidence for a significant difference in the efficiency of complementation by MIT1 and MIT2 (Appendix A).

To explore MIT1 and MIT2 function in Arabidopsis, we identified two and three T-DNA insertion mutant lines for *MIT1* and *MIT2*, respectively (Figure 1A, Appendix A). The T-DNA insertion is located in exon 1 for both *mit1* mutants, causing an interruption in gene expression 200 bp and 53 bp downstream of the start codon in *mit1-1* and *mit1-2*, respectively. In *mit2-1* and *mit2-3* mutants, the T-DNA is located in the intron (419 and 500 bp downstream of the 5′ splice site), while *mit2-2* contains an insertion in the first exon (239 bp downstream of the start codon). We analyzed the progeny of selfed heterozygous *mit1-1*, *mit1-2*, and *mit2-1* plants and found that the progeny did not deviate significantly from 1:2:1 (wild type: heterozygous: homozygous). Furthermore, homozygous mutant plants for each of the *mit1* and *mit2* alleles did not show any phenotypic alteration when compared with wild type plants (data not shown).

Next, RT-PCR analysis of *MIT1* and *MIT2* expression was carried out to ascertain that the homozygous mutant plants obtained for all five mutants (Figure 1B) were truly null mutants (Figure 1C). Results show clearly that *mit1-1*, *mit1-2*, and *mit2-2* plants are knockout mutants. Unexpectedly, *mit2-1* and *mit2-3* accumulate *MIT2* transcript and thus are not knockout mutants. Sequencing of the two *MIT2* RT-PCR products obtained from *mit2-1* RNA demonstrated that the intron is correctly spliced (Appendix A). However, the *MIT2* transcript level as determined by RT-qPCR is significantly decreased (Appendix A), confirming that both *mit2-1* and likely *mit2-3* are knockdown mutants.

Given that *mit1-1*, *mit1-2*, and *mit2-2* are knockout mutants, their normal growth showed that neither *MIT1* nor *MIT2* are essential per se and that they may be redundant. These results led us to perform *mit1* x *mit2* crosses.

### 2.1. MIT Function Is Essential in Arabidopsis

Given that *mit1-1*, *mit1-2*, and *mit2-2* are knockout mutants, their normal growth showed that neither *MIT1* nor *MIT2* are essential per se. To determine whether MIT function is essential, we crossed the knockout mutants *mit1-1* and *mit2-2*. F2 seeds from selfed double heterozygous plants (*MIT1mit1-1 MIT2mit2-2*) were sown on 0.5X MS plates, and a plant carrying three mutated alleles was identified (*MIT1mit1-1 mit2-2mit2-2*). Visual inspection of F3 seeds in three siliques from this selfed plant showed that they contain 77.5 ± 7.0% normal seeds and 22.5 ± 7.0% aborted seeds. Furthermore, when F3 seeds were allowed to develop on soil under iron-sufficient conditions, we were unable to identify a plant carrying a double homozygous mutation (65 plants analyzed). Altogether, these results confirm that MIT function is essential, that the absence of MIT1 and MIT2 is embryo-lethal, and that *MIT1* and *MIT2* genes are redundant.

### 2.2. Crosses Using mit1-1 and mit2-1 Alleles Show Segregation Defects and Produce Abnormal Seeds

Next, we crossed homozygous *mit1-1* plants with homozygous *mit2-1* plants. One hundred and seventy-seven F2 plants grown from the seeds of three selfed F1 double heterozygous plants (genotype *MIT1mit1 MIT2mit2*) were genotyped (Appendix A). No double homozygous mutants were identified, and we noted a bias against plants homozygous for *mit1* and heterozygous for *mit2* (4 plants *mit1-1mit1-1 MIT2mit2-1*) that is apparent but not observed for plants heterozygous for *mit1* and homozygous for *mit2* (22 plants *mit1-1MIT1 mit2-1mit2-1*). This may be due to *mit1-1* being a knockout mutation and *mit2-1* being a knockdown mutation (see below). The plants carrying three mutated alleles did not show visible phenotypic alterations when compared with wild type plants, at least under standard growth conditions.

Then F3 seeds from selfed F2 plants carrying three mutated alleles and one wild type allele (*MIT1* or *MIT2*) were sown directly on the soil, and the grown plants were genotyped. Again, no double homozygous mutant plants were obtained (73 and 65 plants analyzed, Table 1). These results suggest that MIT function is essential and that the *MIT1* and *MIT2* genes are redundant. Furthermore, instead of the expected ratio of 2:1 for heterozygous: wild type plants, *MIT1mit1*:*MIT1MIT1* in the *mit2mit2* background, and *MIT2mit2*:*MIT2MIT2* in the *mit1mit1* background, ratios of 0.8 and 1.1 were observed. These ratios suggest a gametophytic defect, i.e., a defect in gametes carrying only mutated alleles of the mitochondrial iron transporters (*mit1mit2* gametes).

Visual inspection of F4 seeds from F3 plants carrying three mutated alleles showed that in addition to “normal” seeds that resemble those from wild type plants, plants carrying three mutated alleles produced seeds with altered phenotypes: (i) smaller, irregular seeds (“abnormal” seeds), and (ii) shrunken, collapsed seeds (“aborted” seeds) (Figure 2A). Sets of normal, abnormal, and aborted seeds per silique were quantitatively determined using four *MIT1mit1 mit2mit2* plants, four *mit1mit1 MIT2mit2* plants, and two wild type plants as controls (Figure 2B). In siliques from plants with three mutated alleles, 26% (plants with one *MIT1* wild type allele) and 20% (plants with one *MIT2* wild type allele) of total seeds were considered “abnormal”. Only a minor proportion of seeds were aborted (7% and 9%, respectively). There are no significant differences in the total number of seeds per silique between wild type plants and plants with three mutated alleles, nor are there significant numbers of non-fertilized ovules. Furthermore, these numbers were evenly distributed along the inflorescence axis, indicating that seed phenotypes are not due to flower heterogeneity (Appendix A).

When these “abnormal” seeds were sown on plates containing half-concentrated MS medium, almost all were able to germinate (Table 2) and expand their green cotyledons, although at a lower rate than “normal” and wild type seeds (Appendix A). However, only around 50% of seedlings were established with true leaves and elongated roots (Table 2). A significant proportion of these abnormal seeds possessed embryos with three cotyledons (Table 2, Figure 3). We were able to identify double homozygous mutants (*mit1-1mit1-1 mit2-1mit2-1*) among plants grown from “abnormal” seeds, in addition to plants with three mutated alleles (Figure 4). Furthermore, there is a strong correlation between growth rate and genotype: all double homozygous mutants presented a delayed growth rate when compared with plants with either one or two wild type *MIT1* alleles. These results showed that seed morphology was not a clear-cut criterion to identify genotype, and, alternatively, that double homozygous mutants are viable. This unexpected result led us to determine that *mit2-1* is not a knockout mutant (see above, Figure 1C) and that *MIT2* is expressed at a lower level (12.2%) in the double homozygous mutants (Appendix A).

Five individual siliques from *MIT1mit1 mit2mit2* plants were used in two experiments to genotype all seedlings (established or not) grown from normal and abnormal seeds: 16.0 ± 3.5% of total seeds were double homozygous mutants (forty-three out of two hundred and sixty-four total seeds in the five siliques), and 27.7 ± 8.9% of the double homozygous mutants possessed three cotyledons (eleven plants out of forty-three). It is important to point out that all genotyped 3-cotyledon plants in this and other experiments were double homozygous mutants.

### 2.3. Growth of Double Homozygous Mutant Plants Is Severely Affected

Viable double homozygous *mit1-1mit1-1 mit2-1mit2-1* plants were easily identified by their severe phenotype. First, as already mentioned, germination and early post-germinative growth were slower than those of wild type plants and plants with three mutated alleles (Figure 4, Appendix A). For instance, in one experiment, at 13 days, when all Col0 seedlings (90 out of 90) were at least at stage 1.02 on day 13 according to Boyes et al. (2001) [26], only one out of 59 (1.9%) of the double homozygous mutant seedlings attained this stage. Three weeks after germination, only 1.7% (1/59) and 10% (6/59) of these seedlings were at stages 1.06 and 1.04, respectively.

Growth of double homozygous mutant plants on soil was severely affected throughout the entire life cycle, and senescence was delayed by 1.5–2 months (Appendix A). Drastic reduction of MIT expression has pleiotropic effects on double homozygous plants (Figure 5), including pinoid stems (Figure 5A,B) similar to those observed for mutants of auxin efflux carriers (PIN) (39 out of 46 plants, 85%), stems terminated at either cauline leaves (25 plants, 54%; Figure 5B), a unique flower (21 plants, 46%; Figure 5C), or multiple floral buds and cauline leaves (18 plants, 39%; Figure 5D). Furthermore, phyllotaxis in the appearance of cauline leaves was altered: there were a higher number of these leaves in some stems of 15 plants (33%, Figure 5E), their position was less regular (Figure 5D–F), and in some cases three cauline leaves were found at the same position (in seven plants, fifteen percent, Figure 5F). Additionally, some enlarged stems were found in five plants (eleven percent of the plants), as if two stems had been fused (Figure 5G, also visible in the plant shown in Figure 5F).

All double mutant plants showed, alongside some normal flowers, flowers with all their structures (sepals, petals, anthers, and pistils) altered (Figure 6). This resulted in seventeen out of forty-six plants (thirty-seven percent) being unable to give seeds, and the remaining plants showed a reduced seed set (less than fifty seeds in one to six siliques for seventeen plants, between fifty and two hundred and fifty seeds in six to sixteen siliques for ten plants, and more than six hundred seeds for two plants with at least eighty siliques).

### 2.4. The Next Generation of Double Homozygous mit1-1 mit2-1 Mutant Plants Showed a Normal Phenotype

When seeds obtained from *mit1-1mit1-1 mit2-1mit2-1* plants were sown, almost all germinated (97.2 ± 5.0%), and plant establishment was variable (55.5 ± 17.9%). Most importantly, plant growth was similar to that of wild type plants; for instance, all established seedlings were at stage 1.0 at 7 days and at stage 1.02–1.03 at 2 weeks, and this similarity extends to vegetative and reproductive growth. From now on, these plants have been designated as “compensated” double homozygous mutant plants. Their genotype was verified by PCR to be *mit1-1mit1-1 mit2-1mit2-1*.

This intriguing result led us to analyze *MIT2* expression by RT-qPCR in these “compensated” double homozygous mutant plants and compare it with that observed in plants showing an affected phenotype (the first generation of double homozygous mutant plants, arising from seeds obtained from plants carrying three mutated alleles) (Figure 7). Interestingly, *MIT2* expression is significantly higher in compensated plants (60.9% that of wild type plants) compared with affected plants (11.0–18.5% that of wild type plants).

These results are consistent with the view that increased splicing of the *MIT2* intron containing the T-DNA is responsible for phenotypic recovery of the double homozygous mutant plants and may be related to a relatively recently described phenomenon called “T-DNA suppression” (see Discussion).

### 2.5. Mitochondrial Stress Markers UPOX and MSM1 Are Upregulated in the First Generation of mit1-1 mit2-1 Double Mutant Plants

Marker genes for the mitochondrial response to stress have been identified [27], and they include the genes encoding the mitochondrial proteins alternative oxidase 1A and UPOX (upregulated by oxidative stress) [28,29]. On the other hand, Van Aken and Whelan (2012) [30] were able to identify marker genes that respond to mitochondrial and chloroplast dysfunction (e.g., *UPOX*) or are specific for mitochondrial dysfunction (e.g., *MSM1*, for Mitochondrial Stress Marker 1, also designated *At12cys-2*, [31]. We evaluated the expression of these three genes, *AOX1a*, *UPOX*, and *MSM1*, and found that *UPOX* and *MSM1* are significantly upregulated in the first generation of double homozygous mutant plants (*mit1-1mit1-1 mit2-1mit2-1*) but return to wild type levels in the “compensated” second generation plants (Figure 8). In contrast, *AOX1a* transcript levels were not significantly altered in any genotype. These results suggest some degree of mitochondrial perturbation in the first generation of double homozygous mutant plants with a drastic reduction of *MIT* expression.

Unfortunately, we were unable to purify mitochondria from first-generation double homozygous mutant plants (*mit1-1mit1-1 mit2-1mit2-1*) with an affected phenotype. To do this, it would be necessary to grow plants with three mutated alleles, collect seeds, manually separate “abnormal” seeds, and grow plants from these seeds, thus, it was unfeasible to obtain enough biological material. Thus, mitochondria were purified from “compensated” double homozygous mutants (*mit1-1mit1-1 mit2-1mit2-1*) and wild type seedlings as described in Appendix A.

Targeted proteomic analysis was performed on four biological replicates of both compensated and wild type mitochondria by Selective Reaction Monitoring (SRM) mass spectrometry. In this way, more than one hundred proteins (listed in Appendix A) were quantified, allowing a focused dissection of responses in the TCA cycle, electron transport chain, mitochondrial localized iron-related proteins, and MIT1/2 proteins (Appendix A). Significant differences in protein levels between compensated and wild type plants were found only for MIT1 and MIT2. The specific quantified peptide for MIT1 was found in wild type cells but was below the limit of detection in mitochondria from “compensated” double homozygous plants (as expected for a knockout mutation). The specific peptide for MIT2 in compensated plants was 30.9 ± 12.4 (SD)% the level found in wild type mitochondria, confirming that *mit2-1* is a knockdown allele.

Despite this large reduction in MIT abundance (absence of MIT1 and 30% of MIT2), no differences were observed in the oxygen consumption rate of isolated mitochondria (Appendix A). Furthermore, no differences were detected in the abundance or native size of respiratory complexes or in complex I activity when analyzed by BN gel electrophoresis (Appendix A). Complex I activity was assessed as it is the respiratory complex having the highest number of iron ions in its structure, present as iron-sulfur centers. These results show that, at least when plants are grown under standard conditions, mitochondria with a drastic reduction in MIT are not functionally impaired.

### 2.6. Double Homozygous Mutant Plants Misregulate Genes Involved in Iron Uptake, Root Development, and Stress-Related Response

Since RNA-seq analysis required less biological material, we were able to perform this analysis with the first generation of double homozygous mutant plants (*mit1-1mit1-1 mit2-1mit2-1*), which showed an affected phenotype, and we compare this transcriptome with that of “compensated” plants having the same genotype and that of wild type plants. Total RNA was prepared from three biological replicates of 18 day old wild type seedlings and compensated double homozygous mutant seedlings. For double homozygous plants of the first generation (thus presenting a severe phenotype), 27 day old seedlings were considered in order to compare plants at the same developmental stage, in this case, 1.04 of Boyes et al. (2001) [26]. Poly A-enriched RNA fractions were employed to construct libraries for Illumina sequencing (see Methods).

Differentially expressed genes between genotypes were identified (padj < 0.05, log_2_ [fold change] > 1 in any condition, LRT test) and grouped in two clusters by k-means (Figure 9). Interestingly, no significant differences (except for *MIT* expression) were found between wild type plants and compensated plants (Wald test, padj = 3.08 × 10^−18^, log_2_ fold change = −4.2), further supporting the conclusion that partial expression (around 30%) of one of the two *MIT* genes is sufficient for normal plant growth and development. In cluster 1 (408 genes, listed in Appendix A), expression is downregulated in double homozygous mutant plants (1st generation) when compared with wild type and compensated plants. In cluster 2 (360 genes, listed in Appendix A), higher levels of transcripts are found in the double homozygous mutant plants (1st generation).

Enrichment of GO terms (biological processes) was analyzed in both clusters (Appendix A).

Cluster 1 shows an overrepresentation of genes belonging to the Gene Ontology annotation categories of iron ion transport (GO:0006826; FDR 8.6 × 10^−4^, 14.7 fold enrichment) and coumarin metabolic process (GO: 0009804; FDR 1.4 × 10^−4^, 38.1 fold enrichment); coumarins being involved in iron chelation in the rhizosphere for incorporation into the plant [32,33,34]. For instance, *FRO2* (At1g01580, ferric reduction oxidase 2), *IRT1* (At4g19690, iron-regulated transporter 1), *IRT2* (At4g19680, iron-regulated transporter 2), *FIT1/FRU/bHLH29* (At2g28160, FER-like iron deficiency induced transcription factor), *IREG2/FPN2* (At5g03570, iron-regulated transporter 2, ferroportin 2), *BTSL2* (At1g74770, zinc finger BRUTUS-like protein 2), and *NAS2* (At5g56080, nicotianamine synthase 2), are in cluster 1. The downregulation of *FRO2* and *IRT1* was also independently verified by RT-qPCR (Figure 10). Other relevant overrepresented biological processes in cluster 1 are related to growth processes, including root hair elongation, plant-type cell wall modification, and unidimensional cell growth, which are consistent with the growth deficiency phenotypes presented in the double homozygous mutants (Appendix A).

Cluster 2, containing genes with increased expression relative to wild type and compensated plants, shows an overrepresentation of biological processes that can be related to plant defense, including different terms related to jasmonic acid (regulation of jasmonic acid-mediated signaling pathways, jasmonic acid metabolic process, response to jasmonic acid), regulation of defense response, response to other organisms, and glucosinolate metabolic process. As well, processes related to wounding and response to water deprivation were found in cluster 2. This suggests that double homozygous plants have a basal activation of stress-related responses, which might impact plant growth in these plants.

## 3. Discussion

Arabidopsis *MIT1* and *MIT2* encode proteins highly similar to rice MIT and with significant similarity to yeast mitochondrial iron transporters MRS3 and MRS4 (38% identity). MIT1 and MIT2, with their own targeting peptides (Appendix A) or that of MRS3 [19], were able to complement the defect of the yeast Δ*mrs3*Δ*mrs4* mutant, demonstrating that they function as high-affinity iron transporters in yeast mitochondria and likely also in plant mitochondria.

The rice MIT function is essential, since the absence of its unique MIT encoding gene is embryo-lethal [18]. In contrast, in Arabidopsis, the two genes *MIT1* and *MIT2* appear to be redundant since individual mutants, including the knockout mutants *mit1-1* and *mit2-2*, were indistinguishable from wild type plants. Like in rice, MIT function is essential in Arabidopsis since, when crossing these two null mutants, no double homozygous plants could be obtained, and almost 25% of the seeds from plants with three mutated alleles (*MIT1mit1-1 mit2-2mit2-2*) aborted, as expected for a Mendelian segregation.

### 3.1. Phenotypic Alterations of the mit1-1 mit2-1 Double Homozygous Mutants

When crossing the knockout *mit1-1* mutant with the knockdown *mit2-1* mutant, we were able to obtain double homozygous mutant plants from the so-called “abnormal” seeds plated on MS x 0.5 (Figure 2 and Figure 4). The 1st generation of these plants expressed only low levels of *MIT2* (10–20%, Figure 7 and Appendix A) and showed striking phenotypes, highlighting the importance of MIT function. Pleiotropic defects include polycotyly (3-cotyledon embryos, Figure 3 and Table 2), retarded germination and early post-germinative growth with reduced establishment (Appendix A, Figure 4, Table 2), delayed and altered reproductive development, including *pin* stems, abnormalities in phyllotaxy and in all organs of the flowers (Figure 5 and Figure 6), and reduced seed set.

Some of these phenotypes are clearly indicative of defects in auxin signaling. For instance, it is well known that polar auxin transport is involved in cotyledon emergence [35,36], and both *pin* and *pid* mutants show polycotyly [37,38,39,40]. However, whereas the presence of one cotyledon is more frequent in *pin1* mutants (auxin efflux carrier, PINFORMED1), mutants in the ser/thr protein kinase PID (PINOID), which phosphorylates PIN1, show a higher frequency of 3-cotyledon embryos and thus resemble double homozygous *mit1-1 mit2-1* embryos. In all these cases, phenotype penetrance is incomplete. Furthermore, striking similarities are found between *pin*, *pid*, and *mit1-1mit2-1* mutants during reproductive development: inflorescence stems without flowers and cauline leaves (*pin* stems), alterations in cauline leaves’ phyllotaxis, abnormal flowers, and reduced fertility (Figure 5 and Figure 6) [37,41,42].

Although somewhat surprising at first sight, an interplay between mitochondrial function and auxin signaling is well documented [43,44,45,46]. The exact mechanisms linking, in each case, mitochondrial dysfunction and auxin signaling are not known, but several competing or complementary hypotheses have been proposed based on the deep connections between auxin synthesis, conjugation, and post-translational regulation of auxin signaling pathways [47]. These assembled examples and our results suggest that at least some of the phenotypes observed in the *mit1-1mit2-1* double homozygous mutants are mediated by defects in auxin homeostasis.

Alternatively, phenotypes such as reduced plant establishment, delayed germination, and slow early post-germinative growth are characteristic of mitochondrial deficiency. Mitochondria are expected to play a crucial role at these early stages, supplying energy and carbon skeletons for growth, and a role for respiratory complexes I, II, and IV has been described e.g., [48,49,50,51,52,53]. Unfortunately, as already mentioned, we were unable to obtain mitochondria from these seedlings to assess mitochondrial function, and only indirect evidence for mitochondrial dysfunction, i.e., higher transcript levels of *MSM1* and *UPOX*, was documented (Figure 8). However, Bashir et al. (2011) [18] characterized a knockdown mutant of the unique rice *MIT* gene (*mit-2*, T-DNA insertion in the gene promoter), which displays a less severe phenotype, allowing mitochondrial preparation from homozygous mutant plants. Those rice mitochondria contain less Fe and have less aconitase (an iron-sulfur protein) activity, thus supporting mitochondrial dysfunction due to iron deficiency, also in our Arabidopsis double mutant.

### 3.2. Phenotypic Recovery of Double Homozygous mit1-1 mit2-1 in Next Generations

When the few seeds obtained from the affected *mit1-1mit1-1 mit2-1mit2-1* plants were sown, plant growth was similar to that of wild type plants. In these “compensated” double homozygous mutant plants, *MIT2* expression is enhanced with respect to parent plants (Figure 7), and targeted proteomic analysis of purified mitochondria showed they differ significantly from wild type mitochondria only in MIT1 (undetectable) and MIT2 (30%) content (Appendix A). Furthermore, no differences were observed in mitochondrial respiratory rate, respiratory complex size and abundance, or complex I activity. Thus, increased splicing of the *MIT2* intron containing the T-DNA is likely responsible for the phenotypic recovery of the double homozygous mutant plants. In the past years, a phenomenon called “T-DNA suppression” has been described, occurring when crossing two mutants with similar T-DNA insertions (e.g., two SALK lines). At least one of the T-DNA insertions must be intronic, and the “suppressed” phenotype is then caused by the T-DNA [54,55,56,57]. Although the mechanism is not well known, T-DNA hypermethylation and heterochromatinization are necessary, and the RdDM (RNA-dependent DNA methylation) pathway is involved [57]. In our experiments, the altered phenotypes observed in the first generation of double homozygous mutant plants are “suppressed” by an increase in *MIT2* intron splicing.

### 3.3. Transcriptome of Double Homozygous mit1-1 mit2-1 Mutant Plants

No significant differences (except for *MIT* expression) were found between wild type plants and “compensated” *mit1-1mit2-1* double homozygous mutant plants, supporting the conclusion that partial expression (around 30%) of MIT2 is sufficient for normal plant growth and development, at least under standard growth conditions. In contrast, in the first generation of *mit1-1mit2-1* double homozygous mutant plants, which expressed 10–20% of *MIT2* (at the transcript level), 408 genes are down-regulated (cluster 1) and 360 genes are upregulated (cluster 2) compared with wild type and compensated plants (Figure 9, Appendix A).

Interestingly, the data suggest downregulation of the iron acquisition system. Besides *FRO2*, *IRT1* and *IRT2*, genes encoding either proteins involved in coumarin biosynthesis (F6′H1, feruloyl CoA ortho-hydroxylase 1; S8H, Scopoletin 8-hydroxylase; CYP82C4, fraxetin 5-hydroxylase) or coumarins export (ABCG37/PDR9) are included in cluster 1. These genes are part of the Fe deficiency response [58] and are regulated by the master transcription factor FIT1/FRU (Fe-deficiency-induced transcription factor), which is also found in cluster 1 and known to control expression of additional cluster 1 genes involved in Fe homeostasis: *IREG2*, iron-regulated transporter 2/ferroportin 2; *BTSL2*, E3 ubiquitin ligase BRUTUS-like protein 2; and *MTPA2*, metal tolerance protein A2. Therefore, the response to a deficiency in iron uptake by mitochondria may be opposite to the response observed under Fe deficiency reviewed in [34,58] and reminiscent of the root Fe exclusion strategy described in rice [59].

Other enriched GO categories in cluster 1 related to the root system may be indirectly relevant to iron acquisition (Appendix A). Given the role of the root epidermis, and in particular root hairs, in water and nutrient uptake [60,61,62], future work will be necessary to analyze root development in the mutant plants.

Relevant enriched GO terms in cluster 2 (upregulated in mutant plants) may be involved in the observed pleiotropic phenotypes (Appendix A), for instance plant organ formation, anatomical structure formation involved in morphogenesis, flower development, and floral organ development. Furthermore, the enriched GO categories “production of siRNA involved in gene silencing by small RNA”, which has three RNA-dependent RNA polymerases (RDR1, RDR2, and RDR3), and “heterochromatin assembly”, which contains the same three RDR, nucleolin 2, and two chromatin remodeling factors (chr31/SNF2 domain-containing protein CLASSY 3 and chr42/SNF2 domain-containing protein CLASSY 2), may be relevant to explain the T-DNA suppression phenomenon discussed above.

Although enriched GO terms related to auxin were not found, careful examination of the genes in both clusters (Appendix A) highlighted a number of genes related to either auxin metabolism, transport, signaling, or response (9 in cluster 1 and 13 in cluster 2). For instance, PIN2 (At5g57090), ABCG37/PDR9 (At3g53480), ABCB11 (At1g02520), ERULUS (At5g61350), MYB93 (At1g34670), and YUCCA3 (At1g04610) are downregulated in mutant plants (cluster 1). Other genes were found to be upregulated (cluster 2), for example AIL7/PLT7 (At5g65510), LRP1 (At5g12330), SHI (At5g66350), SGR5 (At2g09140), SKP2A (At1g21410), ENP/MACCHI-BOU4/NPY1 (At4g31820), TCP18 (At3g18530), and GH3.5 (At4g27260). Whether these changes are related to the observed phenotypes similar to those of polar auxin transport mutants remains to be explored.

## 4. Materials and Methods

### 4.1. Plant Material and Growth Conditions

All *A. thaliana* plants used were from the Columbia (Col-0) region. Seeds were sown on half-concentrated MS agar medium and stratified for 48 h at 4 °C in the dark. After two weeks in a 16/8 h day/night cycle at 22 °C, seedlings were transferred to soil and grown under long-day conditions (16 h light/8 h dark).

Seeds from five T-DNA insertion mutants were obtained from the ABRC stock center: *mit1-1* (SALK_013388), *mit1-2* (SALK_208340C), *mit2-1* (SALK_096697), *mit2-2* (SAIL_653_B10, CS828300), and *mit2-3* (SALK_095187). For genotyping, DNA was extracted from either 15 day old seedlings or leaves of 4 week old plants and analyzed as described [63]. Primers used to amplify wild type and mutant alleles are described in Figure 1 and Appendix A. To further characterize the T-DNA insertion in *mit2-1*, located in the intron spliced out in spite of its size, both T-DNA/*MIT2* junctions were amplified (Appendix A). All amplified junctions were characterized by DNA sequencing.

Homozygous mutant *mit1-1* plants were crossed with either *mit2-1 or mit2-2* homozygous mutant plants. F1 seeds were germinated, and F1 plants were verified to be double heterozygous plants (genotype *MIT1mit1 MIT2mit2*). F2 seeds obtained from these selfed F1 plants were used to characterize the F2 generation and obtain the next generations (F3, F4, and so on).

### 4.2. Complementation of Mutant Δmrs3Δmrs4 Yeast Cells

The *Δmrs3Δmrs4* strain (*MATa*, *ura3-52*, *leu2-3*, *112*, *trp1-1*, *his3-11*, *ade2-1*, *can1-100*(*oc*), *Δmrs3::kanMax Δmrs4::kanMax*) was kindly provided by Liangtao Li and Diane Ward (Department of Pathology, School of Medicine, University of Utah) and is described in Li and Kaplan (2004) [64]. The constructs containing the *ADH1* promoter, *MIT1* (At2g30160) or *MIT2* (At1g07030) cDNAs, and the *ADH2* terminator were generated using in vivo assembly yeast recombinational cloning [65,66]. *ADH1* promoter and *ADH2* terminator were amplified by PCR using as templates a plasmid kindly provided by Dr. Luis Larrondo and Phusion High-Fidelity DNA polymerase (Thermo Scientific, https://www.thermofisher.com (accessed on 27 February 2023)). For the promoter, either primers 1 and 2 (with an overlap in its 5′ end with the beginning of *MIT2* cds) or primers 1 and 3 (with an overlap in its 5′ end with the beginning of *MIT1* cds) were employed; for the terminator, either primers 8 (with an overlap in its 5′ end with the end of *MIT2* cds) and 9, or primers 10 (with an overlap in its 5′ end with the end of *MIT1* cds) and 9 were used. *MIT1* and *MIT2* cDNAs were obtained by RT-PCR. Total RNA was prepared from 15 day old seedlings with a Spectrum Plant Total RNA Kit (Sigma-Aldrich). cDNAs were synthesized with the Superscript First Strand synthesis system for RT-PCR (Invitrogen, Life Technologies), and PCR amplifications were performed with primers 4 and 5 for *MIT2* and primers 6 and 7 for *MIT1*. PCR products were co-transformed with the linear pRS426 plasmid into the BY4741 yeast strain (*MATa*, *his3*Δ*1*, *leu2*Δ*0*, *LYS2*, *met15*Δ*0*, *ura3*Δ*0*), and circular plasmids obtained from several ura^+^ colonies were transferred to E. coli DH5α. Positive colonies were identified by PCR, plasmids were prepared (AxyPrep Plasmid Miniprep Kit), and construct integrity was verified by DNA sequencing. These plasmids were used to transform the *Δmrs3Δmrs4* strain, and transformants were selected by plating on synthetic defined (SD) medium without uracyl. Complementation assays were performed by growing yeast cells in liquid SD medium (without uracyl) to a DO_600nm_ of 1.5, concentrating five times and plating serial dilutions onto agarose-SD plates with or without 50 µM of the impermeable iron chelator bathophenanthroline disulfonate (BPDS).

### 4.3. Expression Analysis by RT-PCR and RT-qPCR

Total RNA was obtained from frozen seedlings with the TRIzol reagent, treated with DNase I (Promega, http://www.promega.com/ (accessed on 27 February 2023)), and quantified using a Nanodrop spectrophotometer (Thermo Scientific). cDNA synthesis was carried out on 1–2 µg of RNA, using oligodT as a primer and SuperScript II Reverse transcriptase (Thermo Scientific).

To analyze *MIT* expression by RT-PCR in individual mutants, total RNA was prepared from 15 day old seedlings. Then PCR reactions were performed with 1/10 of the cDNA and the following primer pairs (Appendix A): for *MIT1*, either primers 13 and 14 or 15 and 16; for *MIT2*, either primers 17 and 18 or 11 and 12.

For RT-qPCR, total RNA was obtained from seedlings at stages 1.02–1.04 (Boyes et al., 2001). RT-qPCR experiments were performed on 1/10 of the cDNA using the StepOne Plus Real-Time PCR System (Applied Biosystems, http://www.appliedbiosystems.com/ (accessed on 27 February 2023)) according to the manufacturer’s instructions and the Brilliant III Ultra-fast SYBR GREEN QPCR reagents (Agilent). RNA levels were estimated considering the amplification efficiency of each primer pair and normalized relative to either the clathrin adaptor (At4g24550) or the TIP41L (At4g34270) transcripts as internal controls. The primer pairs (Appendix A) used for the clathrin adaptor were At4g24550F and At4g24550R, and for TIP41L, At4g34270F and At4g34270R. Those for *MIT2* are indicated in the figure legends. Primer sequences for *UPOX*, *MSM1*, *AOX1A*, *FRO2*, and *IRT1* are also shown in Appendix A.

### 4.4. Transcriptome Analysis by RNA Sequencing

Total RNA was prepared from seedlings (stage 1.04), using a Spectrum Plant Total RNA Mini Kit (Sigma-Aldrich). Following DNase I (Invitrogen, Life Technologies) treatment, RNA integrity was evaluated by capillary electrophoresis on a Qsep100 Bio-fragment analyzer (BiOptic, https://www.bioptic.com.tw (accessed on 27 February 2023)). Libraries for RNA-Seq were prepared with an Illumina TruSeq Stranded mRNA Kit, quantified by qPCR with a KAPA Library Quantification Kit (Universal) (Roche, https://sequencing.roche.com (accessed on 27 February 2023)), and sequenced using the NextSeq 500 System (Illumina, https://illumina.com (accessed on 27 February 2023)), considering 150 bp paired-end reads. Raw sequences were processed with Trimmomatic v.0.39 ([67]; http://www.usadellab.org (accessed on 27 February 2023)), using the following settings: ILLUMINACLIP:TruSeq3-PE.fa:2:30:10:2:keepBothReads LEADING:30 TRAILING:30 SLIDINGWINDOW:10:30 MINLEN:36. Filtered sequences were mapped to the Arabidopsis genome (The Arabidopsis Information Resource TAIR v.10, www.arabidopsis.org (accessed on 27 February 2023)) using HISAT2 v.2.1.0 with standard settings [68]; http://daehwankimlab.github.io/hisat2/ (accessed on 27 February 2023)). Count tables were generated using the featureCounts function from the Rsubread (v.2.10.4) library from R [69] and the Araport11 GTF gene annotation [70].

Differential gene expression between genotypes was analyzed with the DESeq2 package (v. 1.24; [71], using the likelihood ratio test (LRT). We considered differentially expressed genes those with log_2_ > 1 or log_2_ < 1, and adjusted *p*-value < 0.01. Clustering of differentially expressed genes was performed with the R library heatmap v. 1.0.12 (kmeans_k = 2).

Overrepresentation of Gene Ontology (GO) terms (biological processes) was analyzed for each cluster using the PANTHER Overrepresentation Test tool (http://geneontology.org (accessed on 27 February 2023)) [72].

## Figures and Tables

**Figure 1 plants-12-01176-f001:**
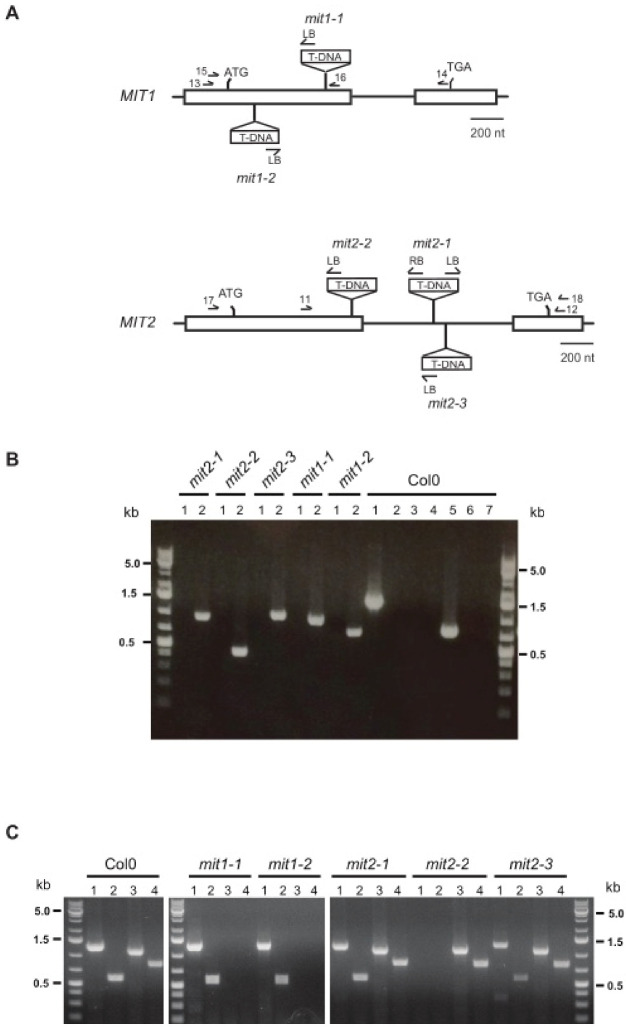
Identification of mit1 and mit2 individual mutants. (**A**) Exon-intron structure of *MIT* genes and T-DNA insertion sites. Exons are represented by boxes, and T-DNA insertion sites in the different mutant lines are shown (T-DNA inserts are not drawn to scale) (details in Appendix A). Horizontal arrows indicate the positions of primers. (**B**) Genotyping of individual *mit1* and *mit2* plants showing they are homozygous mutants. Lanes 1: PCR of the wild type allele (primers 11 and 12 for *MIT2*, 15 and 16 for *MIT1*); lanes 2: amplification of mutant alleles (primers LBb1.3 and 12 for *mit2-1*, 11 and LB1sail for *mit2-2*, 11 and LBb1.3 for *mit2-3*, 15 and LBb1.3 for *mit1-1*, and LBb1.3 and 16 for *mit1-2*). PCR reactions with DNA from wild type Col0 plants were performed with all seven primer pairs (lanes 1 and 5: primers for wild type *MIT2* and *MIT1* alleles; lanes 2 to 4: primers for mutant *mit2-1*, *mit2-2*, and *mit2-3* alleles; lanes 6 and 7: primers for mutant *mit1-1* and *mit2-2* alleles). (**C**) *MIT1* and *MIT2* expression in individual WT, *mit1*, and *mit2* plants. cDNAs from homozygous *mit1-1*, *mit1-2*, *mit2-1*, *mit2-2*, *mit2-3* mutants, and WT plants were amplified (40 cycles) with two primer pairs for each (position indicated in Figure 1A). *MIT* transcript: primers 17 and 18 (lanes 1) or primers 11 and 12 (lanes 2) for *MIT2*, primers 13 and 14 (lanes 3) or primers 15 and 16 (lanes 4) for *MIT1*. Size standards correspond to the GeneRuler 1kb Plus DNA ladder from Thermo Scientific.

**Figure 2 plants-12-01176-f002:**
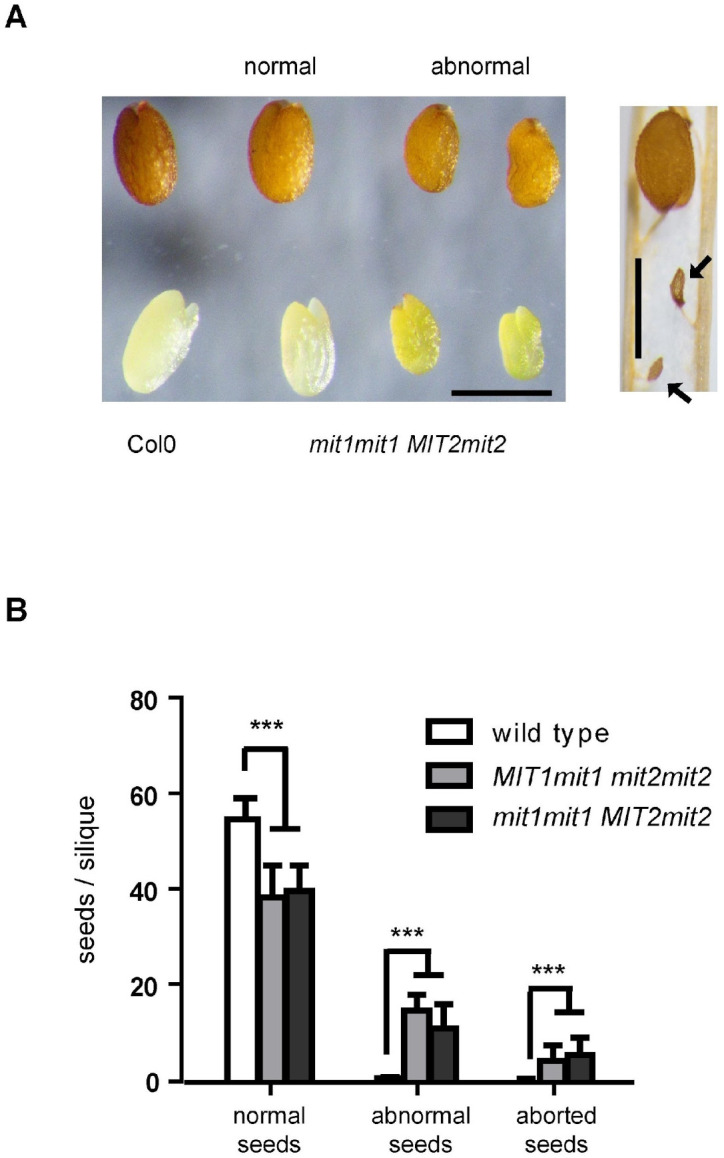
Seed abnormalities in plants carrying three *mit* mutated alleles. (**A**) Mature seeds and manually extracted embryos from wild type Col0 and *mit1mit1 MIT2mit2* plants. “Normal” seeds are those with a wild phenotype; “abnormal” seeds and embryos are somewhat smaller and have an irregular surface. To the right is shown a silique fragment with two shrunken, collapsed seeds, indicated by arrows. Bars = 500 μm. (**B**) Seed (normal, abnormal, and aborted) set per silique was scored along the main inflorescence for four F3 *MIT1mit1 mit2mit2* plants, four F3 *mit1mit1MIT2mit2* plants, and two wild type plants. Twelve siliques per plant (from the 5th to 16th in appearance) were scored. Error bars are SD. Asterisks indicate values that were determined by the *t*-test to be significantly different from wild type (***, *p* < 0.001).

**Figure 3 plants-12-01176-f003:**
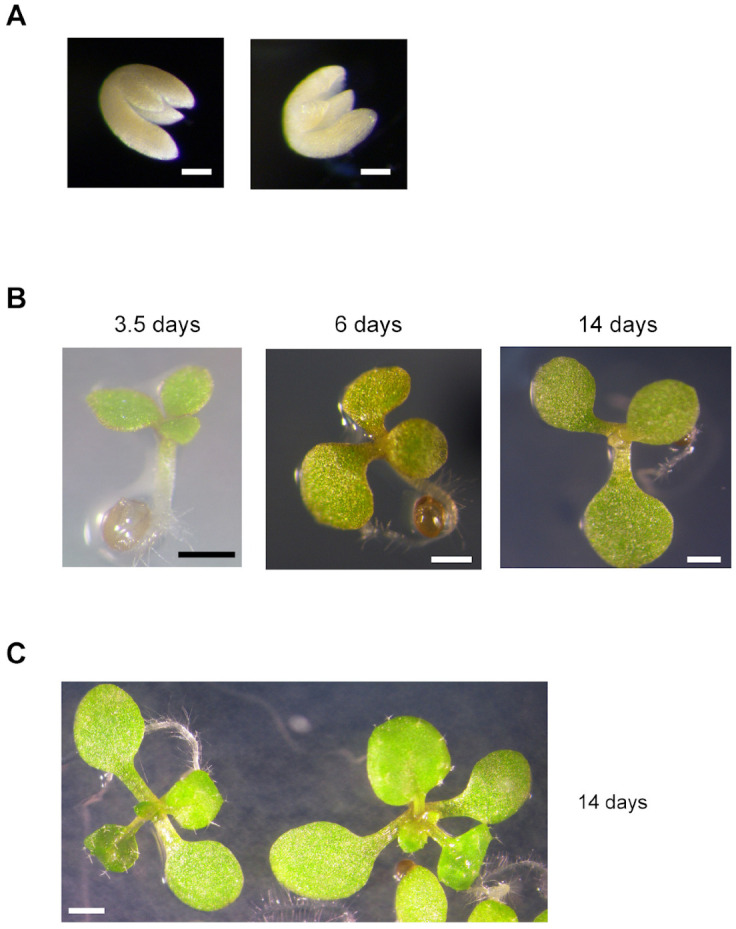
Presence of 3-cotyledon embryos in abnormal seeds. (**A**) Manually extracted mature embryo (two views) from a *MIT1mit1 mit2mit2* plant. (**B**) Abnormal seeds from *mit1mit1 MIT2mit2* plants were sown on 0.5 × MS, stratified for 48 h, and grown for 3.5, 6, and 14 days. The same plant bearing three cotyledons was photographed. (**C**) Control wild type plants grown for 14 days. Bars = 100 μm in (**A**), 500 μm in (**B**), and 1000 μm in (**C**).

**Figure 4 plants-12-01176-f004:**
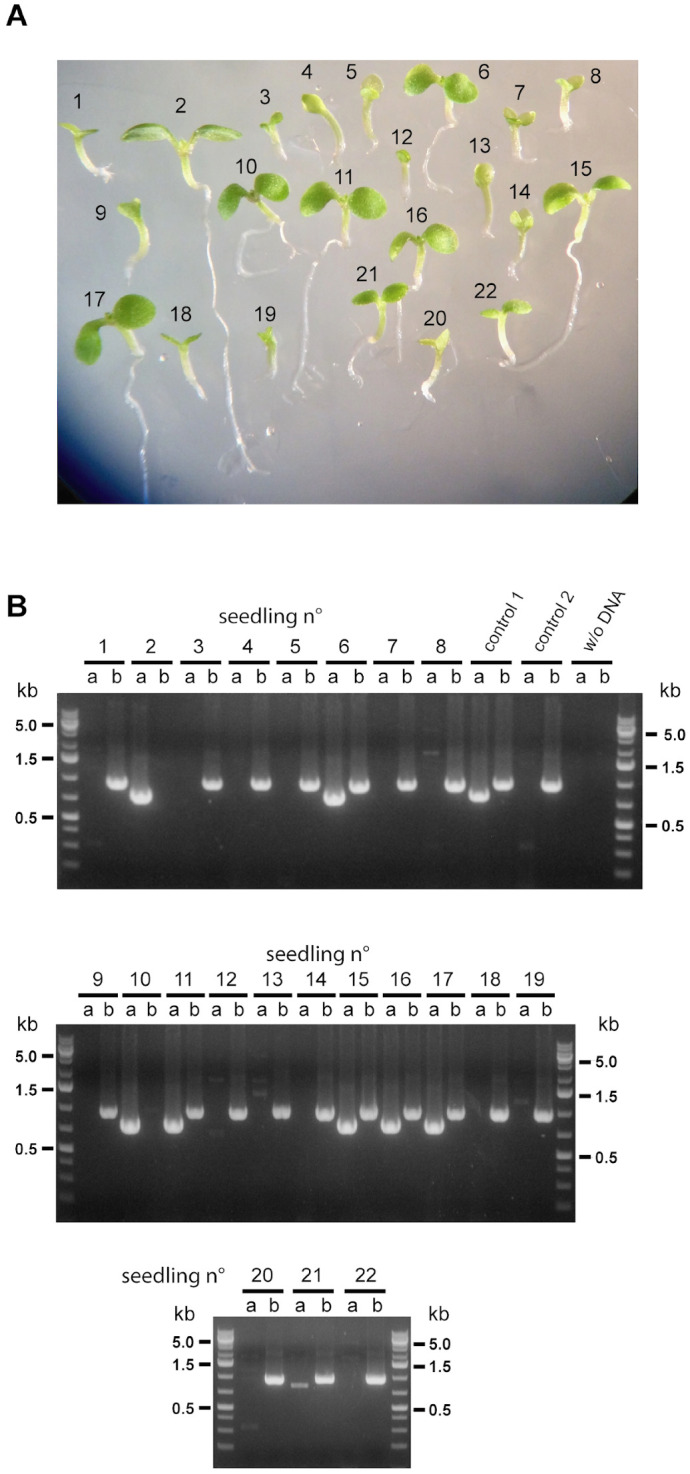
Identification of double homozygous mutant plants. (**A**) Twenty-two abnormal seeds from a *MIT1mit1 mit2mit2* plant were grown on 0.5 × MS and genotyped (**B**) for the presence of the *MIT1* wild type allele (lanes a), using primers 15 and 16, and/or the *mit1-1* mutant allele (lanes b), using primers 15 and LBb1.3. Two plants have the *MIT1MIT1* genotype, six plants have the *MIT1mit1* genotype, and fourteen plants have the *mit1mit1* genotype and are thus double homozygous mutant plants (*mit1mit1 mit2mit2*). Control 1 corresponds to DNA from a previously characterized double heterozygous plant (*MIT1mit1 MIT2mit2*), and control 2 to a homozygous *mit1-1mit1-1* plant. In the figures, n° correspond to number.

**Figure 5 plants-12-01176-f005:**
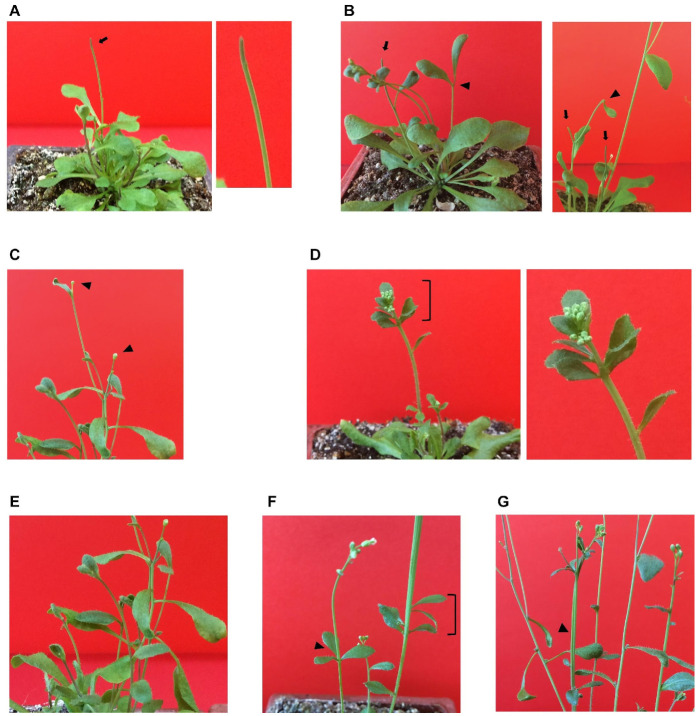
Phenotypic alterations observed in double homozygous plants with drastic reduction of MIT expression. (**A**) A pinoid stem (arrow) with the corresponding enlarged view. (**B**) Two plants with stems terminated at cauline leaves (arrowheads) and which also have pinoid stems (arrows). (**C**) A plant with stems terminated at unique flowers (arrowheads). (**D**) Multiple floral buds and cauline leaves at stem tip. (**E**) Higher number and irregular appearance of cauline leaves. (**F**) Three cauline leaves at the same position (arrowhead). (**G**) Enlarged stem (arrowhead).

**Figure 6 plants-12-01176-f006:**
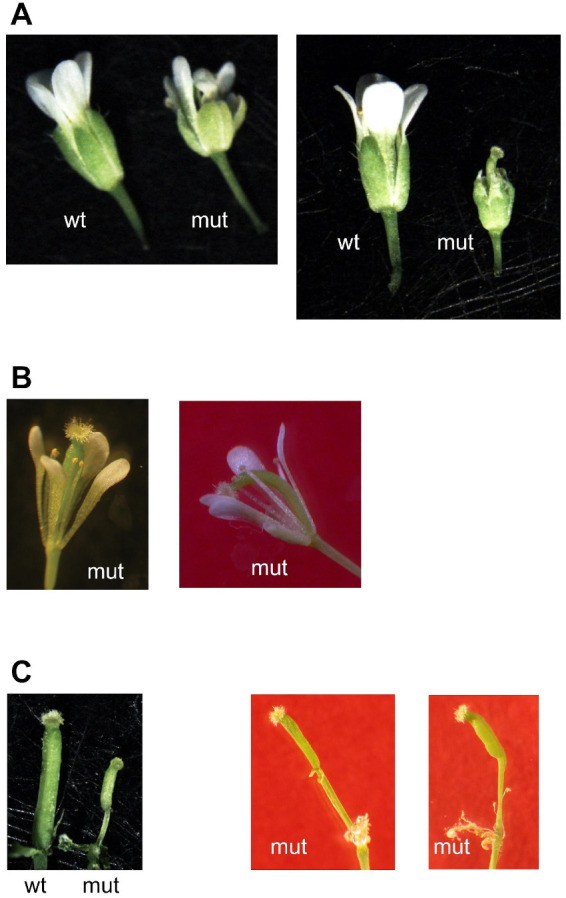
Altered flowers in double homozygous plants with drastic reduction of MIT expression. (**A**) Two different flowers from double homozygous *mit1-1 mit2-1* plants (mut) were compared with wild type (wt) flowers: smaller sepals, smaller petals, lack of anthers (right panel) were visible. (**B**) Two additional mutated flowers from which sepals have been excised: the four petals are heterogeneous in shape, abnormally positioned (right panel), only four and three anthers are present, pistil appears either normal (left panel) or curved (right panel). (**C**) Sepals, petals, and anthers have been excised from wild type and mutated flowers; pistils with a stem-like base are frequent in mutated flowers.

**Figure 7 plants-12-01176-f007:**
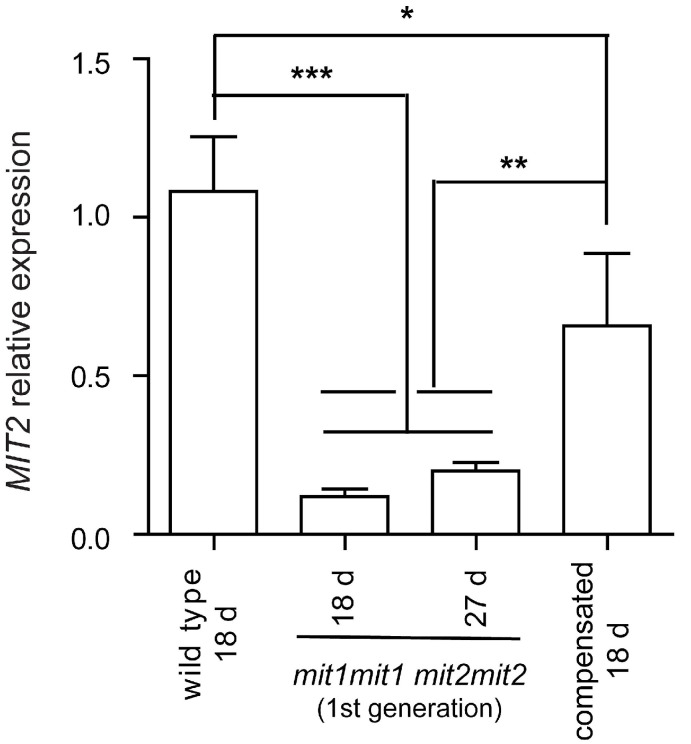
MIT2 expression in double homozygous mutant plants. *MIT2* mature transcript levels were determined by RT-qPCR and normalized to *TIP41-like* transcript levels. Means ± SD of four biological replicates is shown. RNAs were prepared from seedlings at stages 1.03–1.04 [26], i.e., 18 day old seedlings for wild type and “compensated” double homozygous mutant plants (*mit1-1mit1-1 mit2-1mit2-1*) and 27 day old seedlings for the first generation of the double homozygous mutant plants (same developmental stage), and also from 18 day old seedlings from these last plants (stage 1.0). The primers used for *MIT2* were primers 20 (encompassing exon junction) and 21 (Appendix A). Statistically significant differences were determined by one-way ANOVA followed by Tukey’s multiple comparison test (***, *p* < 0.001; **, *p* < 0.01; *, *p* < 0.05).

**Figure 8 plants-12-01176-f008:**
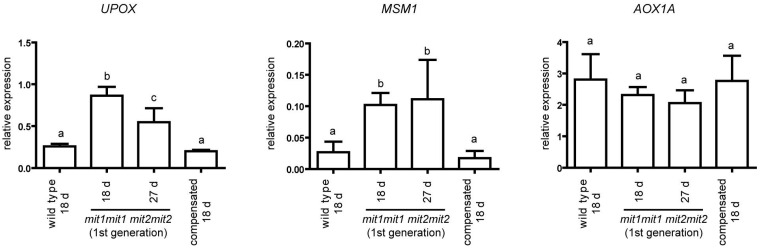
Expression of UPOX, MSM1, and AOX1A in double homozygous mutant plants. Transcript levels were determined by RT-qPCR and normalized to *TIP41-like* transcript level. Means ± SD of four biological replicates is shown. RNAs were prepared from seedlings at stages 1.03–1.04, i.e., 18 day old seedlings for wild type and “compensated” double homozygous mutant plants (*mit1-1mit1-1 mit2-1mit2-1*) and 27 day old seedlings for the first generation of the double homozygous mutant plants (same developmental stage), and also from 18 day old seedlings from these last plants (stage 1.0). Primer sequences are indicated in Appendix A. Statistically significant differences were determined by one-way ANOVA followed by Tukey’s multiple comparison test (same letter indicates no significant differences). For *UPOX*: a-b differences, *p* < 0.001; b-c differences, *p* < 0.01; for MSM1: a-b differences, *p* < 0.05.

**Figure 9 plants-12-01176-f009:**
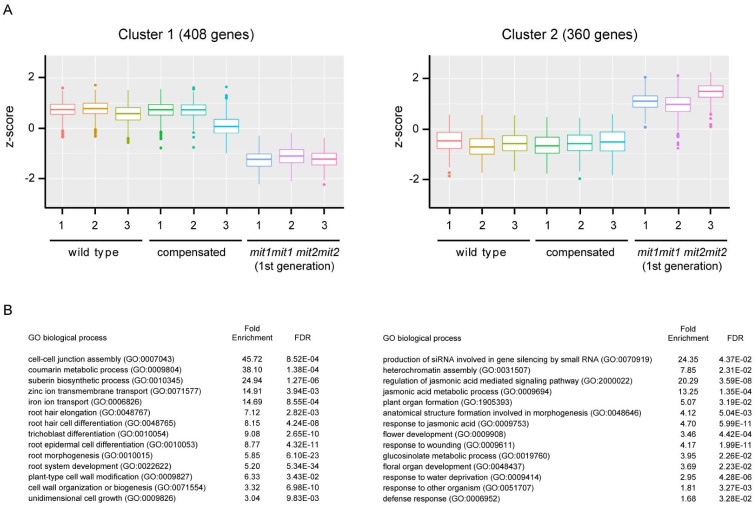
(**A**) K-means clustering of differentially expressed genes. RNAs were prepared from seedlings at the same developmental stage: wild type plants (18 days old), compensated double homozygous mutant plants (18 day old), and 1st generation of double homozygous mutant plants with an affected phenotype (27 day old). RNAseq was performed on three biological replicates (seedlings grown on different days) for each group. Differentially expressed genes (at least two-fold) were identified using DEseq2 software and clustered by k-means. (**B**) Some enriched GO terms (biological processes) in clusters 1 and 2. See Appendix A for more detailed information.

**Figure 10 plants-12-01176-f010:**
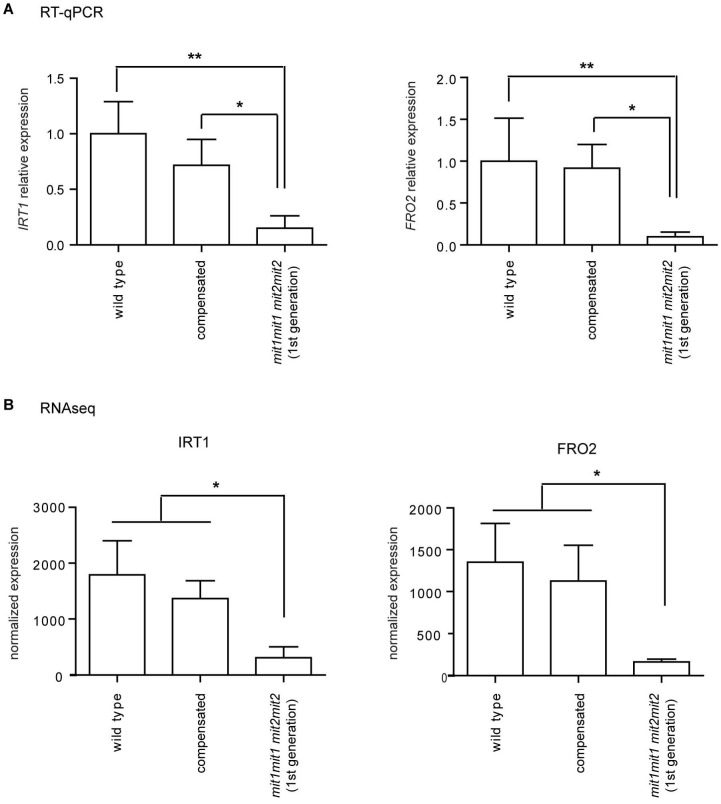
IRT1 and FRO2 expression in double homozygous mutant plants. (**A**) *IRT1* and *FRO2* mature transcript levels were determined by RT-qPCR and normalized to *TIP41-like* transcript levels. RNAs were prepared from seedlings at stages 1.03–1.04 (same developmental stage). Primers sequences are indicated in Appendix A. Means ± SD of three (*IRT1*) or four (*FRO2*) biological replicates are shown. Statistically significant differences were determined by one-way ANOVA followed by Tukey’s multiple comparison test (**, *p* < 0.01; *, *p* < 0.05). (**B**) Normalized expression of *IRT1* and *FRO2* as determined by RNAseq. Means ± SD of the three biological replicates are shown; a one-way ANOVA was performed, followed by Tukey’s multiple comparison test (*, *p* < 0.05).

**Table 1 plants-12-01176-t001:** Segregation analysis in the progeny of F2 plants carrying three mutated alleles.

F2 plant genotype: *MIT1mit1mit2mit2.*			
**Analyzed F3 Plants**	**Genotype of F3 Plants**		**Ratio**
	*MIT1mit1mit2mit2*	*MIT1MIT1mit2mit2*	
73	32	41	0.8
F2 plant genotype: *mit1mit1MIT2mit2.*			
**Analyzed F3 Plants**	**Genotype of F3 Plants**		**Ratio**
	*mit1mit1MIT2mit2*	*mit1mit1MIT2MIT2*	
65	34	31	1.1

F3 seeds obtained from selfed F2 plants carrying three mutated alleles were sown directly on soil, and inheritance of *mit1* and *mit2* alleles were analyzed by genotyping 73 and 65 F3 plants, respectively. Uppercase letters indicate wild type alleles, and lowercase letters indicate mutated alleles. Seedling genotypes were determined by PCR, as described in Methods. Expected ratio for Mendelian inheritance is 2.0.

**Table 2 plants-12-01176-t002:** Characterization of “abnormal” seeds obtained from plants with three mutated *mit* alleles.

	“Abnormal” Seeds From	
	*MIT1mit1mit2mit2*	*mit1mit1MIT2mit2*
% germination	94.4 ± 7.8 ^1^	97.2 ± 2.6 ^2^
% establishment	57.4 ^3^	53.9 ± 11.9 ^4^
% 3-cotyledon embryos	12.1 ± 9.3 ^5^	14.8 ± 8.4 ^6^

Abnormal seeds of *MIT1mit1mit2mit2* and *mitmit1MIT2mit2* plants were sown on 0.5xMS plates, and different parameters evaluated. Germination was recorded as radicle protrusion and establishment as appearance of leaves and root growth. Replicate numbers: ^1^ 5 replicates (20–30 seeds each), ^2^ 11 replicates (30–40 seeds each), ^3^ 1 experiment with 29 seeds, ^4^ 8 replicates (29–37 seeds each), ^5^ 5 replicates (15–33 seeds each, 15 embryos with 3 cotyledons out of 122 total abnormal seeds), ^6^ 10 replicates (30–40 seeds each, 48 3-cotyledon embryos out of 310 total abnormal seeds).

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
