# Peer review of "Growth Developmental Defects of Mitochondrial Iron Transporter 1 and 2 Mutants in Arabidopsis in Iron Sufficient Conditions"

_plants, 2023, doi:10.3390/plants12051176_

Round 1
Reviewer 1 Report
The manuscript:
Growth Developmental Defects of Mitochondrial Iron Transporter 1 and 2 Mutants in Arabidopsis in Iron Sufficient Conditions
by Vargas et al
describes the characterization of mit1mit2 double mutants. The manuscript presents flaws listed below.
-physiological relevance of MIT1 and MIT2 has been already published by Erin Connolly's group , some years ago (Jain et al, 2019). The authors of the present manuscript should have referred to this paper and should have highlighted the real novelties of their work. The production of their mutants and the genetics behind it, constitutes a big part of the manuscript itself, not always clearly presented.
-The manuscript is not always clear in its scientific achievements; indeed, the titles of the paragraphs in the Results instead of being clear take-home messages of the achieved results, are descriptive of the adopted technical approaches. The results should be therefore presented in a clearer and more easily readable way. Technical details should go in supplementary material, speculative parts should go in the discussion in order to make main text simpler and more focused. As an example: the authors cite auxin in the abstract but a clear paragraph with results supporting such conclusion is missing.
-the discussion is also unclear; moreover, it should ideally not contain any references to single picture or tables of the manuscript.
Author Response
Reviewer 1
We thank the reviewer for its suggestions. All the modifications have been highlighted in the new version
The manuscript: Growth Developmental Defects of Mitochondrial Iron Transporter 1 and 2 Mutants in Arabidopsis in Iron Sufficient Conditions by Vargas et al describes the characterization of mit1mit2 double mutants. The manuscript presents flaws listed below.
-physiological relevance of MIT1 and MIT2 has been already published by Erin Connolly's group, some years ago (Jain et al, 2019). The authors of the present manuscript should have referred to this paper and should have highlighted the real novelties of their work.
-Regarding the novelty of our findings:
i) We were able to obtain double homozygous plants when performing the cross (mit1-1x mit2-1) and thus to describe the severe phenotypes observed with such a drastic reduction in MIT expression (not obtained by Jain et al.): three-cotyledon embryos, delayed growth rate, severe phenotypes reminiscent of defects in auxin homeostasis (e.g. pinoid stems), flower abnormalities. Additionally, the cross between two knockout mutants (mit1-1 x mit2-2), i.e. formal proof of MIT essentiality, was not performed in Jain et al.
ii) Compensation phenomenon, likely due to “T-DNA suppression”, is interesting per se, since it strengthens the conclusion that phenotype severity correlates with MIT expression level. And established at the protein level that 30 % of MIT2 (and 0 % of MIT1) is enough for normal growth under iron-sufficient conditions.
iii) The effect of drastic reduction of MIT expression on the transcriptome.
-We included in different parts of the manuscript citations to the Erin Connolly’s research and highlighted the novelties of our findings.
The production of their mutants and the genetics behind it, constitutes a big part of the manuscript itself, not always clearly presented.
-The description of the double mutants obtaining was simplified in the manuscript.
-The manuscript is not always clear in its scientific achievements; indeed, the titles of the paragraphs in the Results instead of being clear take-home messages of the achieved results, are descriptive of the adopted technical approaches. The results should be therefore presented in a clearer and more easily readable way. Technical details should go in supplementary material, speculative parts should go in the discussion in order to make main text simpler and more focused. As an example: the authors cite auxin in the abstract but a clear paragraph with results supporting such conclusion is missing.
-Some titles of the paragraphs in the results sections were changed in order to emphasize the main results obtained.
Title 2.1 “Isolation of mit1-1 x mit2-1, a MIT knockdown mutant” was changed by “Crosses using mit1-1 and mit2-1 alleles show segregation defects and produced abnormal seeds”
Title 2.5 “Analysis of mitochondrial function” was changed by “Mitochondrial stress markers UPOX and MSM1 are upregulated in the first generation of mit1-1 mit2-1 double mutant plants”
Title 2.6 “RNA-seq analysis of gene expression in double homozygous mutant plants” was changed by “Double homozygous mutant plants misregulate genes involved in iron uptake, root development, and stress related response”.
-Several technical details and speculative parts included in the results were transferred to the material and methods and discussion sections.
-We modified the abstract changing “our data suggest that some of the phenotypes observed in Atmit1 Atmit2 double homozygous mutant plants are mediated by defects in auxin homeostasis” by “The phenotypes observed, like pinoid stems and fused cotyledons, in Atmit1 Atmit2 double homozygous mutant plants may suggest defects in auxin homeostasis.”.
-the discussion is also unclear; moreover, it should ideally not contain any references to single picture or tables of the manuscript.
The discussion was modified as the reviewer suggested. Several parts of the discussion were simplified or deleted to avoid redundancy. Additionally, the abstract was modified emphasizing the main results and conclusions of our findings.

Reviewer 2 Report
First, I would like to congratulate authors for presenting well-structured manuscript. The manuscript requires following revisions
Figure 1C: Labeling of lanes in gels are incorrect or confusing. WT for Mit1-1 and -2 is missing
The abstracts need to be precise and summaries main results and conclusion from the study.
Gene expression data (RT-qPCR) for AtMIT1 and AtMIT2 in mutant line is missing. Only RT-PCR is not sufficient
Paragraph: 163-175 - Lacks clarity and information needs to be rephrased.
Authors do not offer any explanation/discussion as to for observation in “Line 486-491” if everything else was similar in rice.
The discussion needs to be precise and how does the segregation data compare with other crops like rice?
Author Response
Reviewer 2
We thank the reviewer for its suggestions. All the modifications have been highlighted in the new version.
First, I would like to congratulate authors for presenting well-structured manuscript. The manuscript requires following revisions:
Figure 1C: Labeling of lanes in gels are incorrect or confusing. WT for Mit1-1 and -2 is missing
We apologize if the Figure 1C was not clear. We modified the Figure 1C in order to avoid confusions.
All the plants used in the RT-PCR analysis are Col0 ecotype, and the composed figure used the same WT to compare the size of RT-PCR products for all mutant alleles.
Also, we modified the legend of the figure 1C as following, “(C) MIT1 and MIT2 expression in individual WT, mit1 and mit2 plants. cDNAs from homozygous, mit1-1, mit1-2 mit2-1, mit2-2, mit2-3 mutant and WT plants were amplified (40 cycles) with two primer pairs for each (position indicated in Figure 1A) MIT transcript: primers 17 and 18 (lanes 1) or primers 11 and 12 (lanes 2) for MIT2, primers 13 and 14 (lanes 3) or primers 15 and 16 (lanes 4) for MIT1. Size standards correspond to the GeneRuler 1kb Plus DNA ladder from Thermo Scientific.”
The abstracts need to be precise and summaries main results and conclusion from the study.
-The abstract was modified emphasizing the main results and conclusions of our findings. All the modifications have been highlighted in the new version.
Gene expression data (RT-qPCR) for AtMIT1 and AtMIT2 in mutant line is missing. Only RT-PCR is not sufficient
-The main point in the characterization of the single mutant plants was to determine if the lines used were null alleles. For this reason, we consider that RT-PCR was sufficient to answer the question. Additionally, in iron sufficient conditions single mutant plants did not show phenotypes and our more quantitative molecular characterization of the gene expression was reserved for the double mutant plants, which were used in furthers analysis. However, RT-qPCR was included in supplementary figure S3 for mit2-1 and mit2-3 mutant plants.
Paragraph: 163-175 - Lacks clarity and information needs to be rephrased.
-The paragraph was simplified in order to keep the relevant information. The speculative phrase about splicing of long introns in plants was deleted.
We simplified the paragraph as following, “Next, RT-PCR analysis of MIT1 and MIT2 expression was carried out to ascertain homozygous mutant plants obtained for all five mutants (Figure 1B) were truly null mutants (Figure 1C). Results show clearly that mit1-1, mit1-2 and mit2-2 plants are knockout mutants. Unexpectedly, mit2-1 and mit2-3 accumulate MIT2 transcript, and thus are not knockout mutants. Sequencing of the two MIT2 RT-PCR products obtained from mit2-1 RNA demonstrated that the intron is correctly spliced (Supplementary Figure S3). However, MIT2 transcript level as determined by RT-qPCR is significantly decreased (Supplementary Figure S6), confirming that both mit2-1 and likely mit2-3 are knockdown mutants”.
Authors do not offer any explanation/discussion as to for observation in “Line 486-491” if everything else was similar in rice. The discussion needs to be precise and how does the segregation data compare with other crops like rice?
We modified the paragraph as following, “The rice MIT function is essential, since the absence of its unique MIT encoding gene is embryo-lethal (Bashir et al., 2011). In contrast, in Arabidopsis the two genes MIT1 and MIT2 appear to be redundant since individual mutants, including the knockout mutants mit1-1 and mit2-2 were indistinguishable from wild type plants. Like in rice, MIT function is essential in Arabidopsis since when crossing these two null mutants no double homozygous plants could be obtained, and almost 25 % of the seeds from plants with three mutated alleles (MIT1mit1-1 mit2-2mit2-2) aborted, as expected for a Mendelian segregation.”

Reviewer 3 Report
Please delete green highlights from text.
Round 2
Reviewer 2 Report
N/A